# Can resistant coral-*Symbiodinium* associations enable coral communities to survive climate change? A study of a site exposed to long-term hot water input

Shashank Keshavmurthy[1], Pei-Jie Meng[2,3], Jih-Terng Wang[4], Chao-Yang Kuo[1,5], Sung-Yin Yang[6], Chia-Min Hsu[1,7], Chai-Hsia Gan[1], Chang-Feng Dai[7] and Chaolun Allen Chen[1,7,8]

[1] Biodiversity Research Center, Academia Sinica, Nangang, Taipei, Taiwan
[2] National Museum of Marine Biology/Aquarium, Checheng, Pingtung, Taiwan
[3] Institute of Marine Biodiversity and Evolution, National Dong Hwa University, Checheng, Pingtung, Taiwan
[4] Institute of Biotechnology, Tajen University of Science and Technology, Pintung, Taiwan
[5] ARC Centre for Coral Reef Studies, James Cook University, Townsville, Australia
[6] University of Ryukyus, Graduate School of Engineering and Science, Okinawa, Japan
[7] Institute of Oceanography, National Taiwan University, Taipei, Taiwan
[8] Taiwan International Graduate Program (TIGP)-Biodiversity, Academia Sinica, Nankang, Taipei, Taiwan

Corresponding author
Chaolun Allen Chen,
cac@gate.sinica.edu.tw

## ABSTRACT

Climate change has led to a decline in the health of corals and coral reefs around the world. Studies have shown that, while some corals can cope with natural and anthropogenic stressors either through resistance mechanisms of coral hosts or through sustainable relationships with *Symbiodinium* clades or types, many coral species cannot. Here, we show that the corals present in a reef in southern Taiwan, and exposed to long-term elevated seawater temperatures due to the presence of a nuclear power plant outlet (NPP OL), are unique in terms of species and associated *Symbiodinium* types. At shallow depths (<3 m), eleven coral genera elsewhere in Kenting predominantly found with *Symbiodinium* types C1 and C3 (stress sensitive) were instead hosting *Symbiodinium* type D1a (stress tolerant) or a mixture of *Symbiodinium* type C1/C3/C21a/C15 and *Symbiodinium* type D1a. Of the 16 coral genera that dominate the local reefs, two that are apparently unable to associate with *Symbiodinium* type D1a are not present at NPP OL at depths of <3 m. Two other genera present at NPP OL and other locations host a specific type of *Symbiodinium* type C15. These data imply that coral assemblages may have the capacity to maintain their presence at the generic level against long-term disturbances such as elevated seawater temperatures by acclimatization through successful association with a stress-tolerant *Symbiodinium* over time. However, at the community level it comes at the cost of some coral genera being lost, suggesting that species unable to associate with a stress-tolerant *Symbiodinium* are likely to become extinct locally and unfavorable shifts in coral communities are likely to occur under the impact of climate change.

## INTRODUCTION

As a result of global change, tropical ocean temperatures are predicted to rise between 1.0 and 3.0 °C by the end of this century (*IPCC, 2013*) in addition to fluctuation resulting from an increase in minimum and maximum temperatures with daily minimum temperatures rising more rapidly than maximums (*Traill et al., 2010*; *Vose, Easterling & Gleason, 2005*). Evidence from research has shown a higher sensitivity to global warming in tropical species since they are exposed to narrower thermal niches (*Kozak & Wiens, 2007*). Coral holobionts (in this paper refers to coral host + zooxanthellae, see *Weber & Medina, 2012*) are known to be living at, or near, their tipping points (*Donelson & Munday, 2012*; *Monaco & Helmuth, 2011*) across a range of different thermal environments (*Ladner, Barshis & Palumbi, 2011*). In the absence of significant mechanisms to resist stress, corals will likely face high levels of regional mortality through time (*Ladner, Barshis & Palumbi, 2011*). For corals, it is now imperative to be able overcome the effects of climate change and continue to survive. The question remains as to whether every coral species can overcome the effects of climate change. It is necessary for the whole coral community in any given location to survive, as survival of only a few species cannot maintain the coral reef community. *Marshall & Baird (2000)* have suggested that the change in community structure and species diversity is a result of the differences among species in their susceptibilities to disturbance. The fact that relatively small excursions of seawater temperature can have large-scale impacts on coral survival indicates that reef-building corals are living close to their upper thermal limit (*Fitt et al., 2001*; *Riegl et al., 2011*). If environmental perturbations exceed the adaptive capacity of corals, it may result in a change in their communities over time such that species that are phenotypically plastic or can adapt genetically through time will become dominant. In a disturbed environment where organisms under constant stress or challenged by increasing stress over time (for example, salinity or temperature increases), the number of individuals exposed to selection will be greater resulting in an overall shift in abundance and a change in composition (*Bell, 2012*).

Over short periods of time (within a single generation), the potential for corals to acclimatize to climate change through phenotypic plasticity or by specific combinations with stress resistant *Symbiodinium* types thorough natural selection may be the over-riding determinant of survival (*Marshall & Baird, 2000*). However, a beneficial association between a coral host and *Symbiodinium* is a rather complex and holistic process because the classification of *Symbiodinium* into functionally distinct evolutionary entities (using alpha-numeric designations equivalent to 'species') has shown them to belong to nine divergent phylogenetic 'clades' (A to I) (*Pochon & Gates, 2010*). Corals associate with either single or multiple *Symbiodinium* clades, and occupy defined ecological niches and roles within and across coral hosts (*LaJeunesse et al., 2010*; *Pochon & Gates, 2010*; *Weber & Medina, 2012*) based on their physiological response, some of which includes photosynthetic efficiency (*Iglesias-Prieto et al., 2004*) and sensitivity or tolerance to heat stress (*Baker, 2003a*; *Baker, 2003b*; *Jones et al., 2008*; *Little, van Oppen & Willis, 2004*; *Rowan, 2004*; *Sampayo et al., 2008*; *Warner et al., 2006*), light adaptation. While spatial pattern in symbiont communities can be explained through identification at the clade

level, the exclusion of intra-cladal differences more often obscures ecological patterns in *Symbiodinium* distribution (*Tonk et al., 2013*). With increasing attention and advanced sequencing methods (*Weber & Medina, 2012*), diverse 'sub species' or 'types' have been discovered within the *Symbiodinium* clades, and various studies have discussed on length about the *Symbiodinium* type physiological differences and their contribution to coral host stress resistance. Similar to the case of *Symbiodinium* clade; geographic locations, local environmental conditions (differences in physical parameters such as temperature, light and turbidity) have been found to define the *Symbiodinium* type associated with coral hosts. For example, corals present in shallow and turbid locations are found to be associated with *Symbiodinium* type D1a (*LaJeunesse et al., 2010*) or the distribution of *Symbiodinium* types according to different light environments (*Farde et al., 2008*). *Thornhill et al. (2006)* have shown that *Orbicella annularis* from Florida Key were associated with *Symbiodinium* type D1a after the 1998 bleaching event and reverted back to being associated with *Symbiodinium* type B10 after 2002. Results from the laboratory experiments have shown difference in tolerance to temperature stress (*Brading et al., 2011*; *Fisher, Malme & Dove, 2012*; *Kramer et al., 2010*; *Wang et al., 2012*). For example, *Symbiodinium* type A1 was found to be more tolerant to temperature stress compared to types C1 and B1 (*Hawkins & Davy, 2012*) and *Symbiodinium* type C15 was more tolerant than type C3 in terms of their photosynthetic efficiency (*Fisher, Malme & Dove, 2012*). With respect to *Symbiodinium* type D1a; studies have shown highest activation energy when subjected to temperature stress in freshly isolated D1a when compared to B1, C1, C3 and C15 types (*Wang et al., 2012*). In a cold temperature stress experiment, *Thornhill et al. (2006)* showed that *Symbiodinium* type B2 displayed rapid and full recovery at 10 °C for 2-week period in their photochemical efficiency compared to types A3, B1 and C2 (*Fisher, Malme & Dove, 2012*). These studies have revealed the importance of *Symbiodinium* types in assisting the coral host resisting mechanisms to stress.

In addition to associating with a resistant *Symbiodinium* clade/type, corals can overcome environmental perturbations, mainly seawater temperature fluctuations, by their ability to shuffle between clades/types depending on the environmental and seasonal condition (*LaJeunesse et al., 2004*; *Chen et al., 2005*; *Thornhill et al., 2006*; *Sampayo et al., 2008*; *LaJeunesse et al., 2010*; *Hsu et al., 2012*; *Keshavmurthy et al., 2012*). It has often been found that corals associating with stress resistant *Symbiodinium* clade D have experienced community changes that resulted in some coral species being favored over others (*Marshall & Baird, 2000*; *Loya et al., 2001*). Studies have shown the possible ability of coral and various *Symbiodinium* combinations to respond to the effects of climate change as a result of a high degree of variation in coral and symbiont thermal tolerance and symbiont community shifts in response to thermal tolerance (*Marshall & McCulloch, 2002*; *Hughes, 2003*; *Baker, 2001*; *Baker et al., 2004*; *Berkelmans & van Oppen, 2006*). Short-term field or laboratory studies that have been performed to understand the acclimatization and/or adaptation potential in corals to climate change (*Meesters et al., 2002*; *Coles & Brown, 2003*; *Ayre & Hughes, 2004*; *Baker et al., 2004*; *Rowan, 2004*; *Richier et al., 2005*; *Berkelmans & van Oppen, 2006*; *Gittenberger & Hoeksema, 2006*; *Shaish, Abelson & Rinkevich, 2007*; *Strychar*

*& Sammarco, 2009*; *Hennige, Smith & Walsh, 2010*; *Barshis et al., 2010*; *Oliver & Palumbi, 2011*; *Bellantuono, Hoegh-Guldberg & Rodriguez-Lanetty, 2011*; *Keshavmurthy et al., 2012*; *Howells et al., 2013*) have their own shortfalls. To understand more comprehensively the capacity of acclimatization and adaptation, a more effective way would be to conduct mesocosm experiments by mimicking different perturbations of climate change, but such experiments are difficult to conduct and suffer from logistical problems. Another way is to find a location that already is in a situation where seawater temperatures are similar to levels predicted for 2050 by IPCC and is also subjected to natural and anthropogenic disturbances over time.

In Kenting, southern Taiwan, there is an area that is influenced by constant hot-water effluent from a nuclear power plant located along the western side of Nanwan Bay that has been operating since 1984. At this nuclear power plant outlet (NPP OL), the average seawater temperature is 2.0–3.0 °C higher than at other coral reef sites in Kenting (*Fan, 1991*; *Pier, 2011*; also see *Keshavmurthy et al., 2012*). Hot water at the nuclear power plant (NPP) site is trapped and flows southwest in Nanwan Bay because of a near-shore current and tides (*Chiou, Cheng & Ou, 1993*). The hot water released in this area has had an impact on the marine ecology within the area of dispersal (*Chiou, Cheng & Ou, 1993*; *Hung, Huang & Shao, 1998*; *Jan et al., 2001*; *Hwang, Tsai & Lee, 2004*). Recent studies on two coral species, *Isopora palifera* and *Platygyra verweyi* (sampling done on a local scale in Kenting, Taiwan), showed a differential trend in associating with *Symbiodinium* type D1a. For example, *P. verweyi*, distributed in shallow waters at depths of 2–4 m in Southern Taiwan, is especially abundant near NPP OL. This species does not show obvious bleaching at NPP OL even though it is constantly exposed to warm water, and is associated with *Symbiodinium* type D1a, whereas it associates with *Symbiodinium* type C3 in cooler waters (*Keshavmurthy et al., 2012*). In the case of *Isopora palifera*, it overcomes the effect of hot water at NPP OL by shuffling its *Symbiodinium* clades seasonally (*Hsu et al., 2012*) and through the presence of exclusive stress-resistant haplotypes of the host (*Hsu et al., 2012*).

By utilizing Kenting and NPP OL as a natural mesocosm, we were able to investigate the present composition of the coral community and associated *Symbidodinium* types in sixteen genera of reef-building corals exposed to elevated seawater temperatures over the previous 26 years. For comparison, we sampled the same sixteen genera from seven adjacent sites that are not under the influence of the nuclear power plant's hot water effluent to see how a long-term environmental perturbation (in this case seawater temperature) affected present day coral-*Symbiodinium* and coral host composition. Due to the lack of historical *Symbiodinium* composition data, and although it is not possible to show the process of acclimatization, we based our study on the hypothesis that the present composition of coral-*Symbiodinium* and coral hosts at the nuclear power plant location is the result of acclimative and/or adaptive phenotypic plasticity during 26 years of exposure to chronically elevated seawater temperatures. This could be due to the plasticity in the coral hosts or to associations with resistant *Symbiodinium* types. By conducting large-scale sampling from 16 coral genera, this study attempts to understand how different coral species in a community might have adjusted or responded to long-term elevated

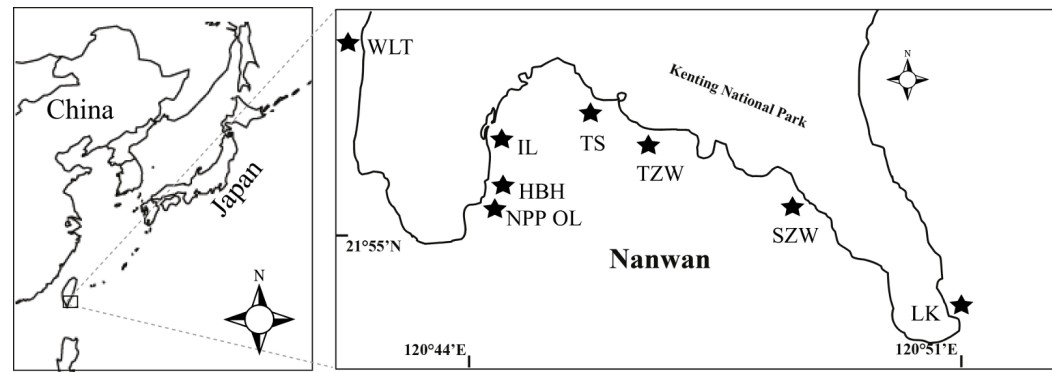

**Figure 1 Map of the study area and sampling locations in Kenting, southern Taiwan.** Eight sampling sites including the nuclear power plant outlet (NPP OL) sites are shown by black stars.

seawater temperature stress. The possible outcomes are that either all corals exposed to chronic seawater temperature stress host only resistant types of *Symbiodinium*, or only those corals that are able to host resistant *Symbiodinium* types can survive. We discuss whether resistance at the genus or species level, if any, towards the long-term warming of seawater temperatures will enable corals at the community level to survive the effects of climate change.

## MATERIAL AND METHODS

### Area description and sampling

Coral samples were obtained from eight sites present in Nanwan, Kenting located at the southern tip of Taiwan (Fig. 1). Of the eight sites, one was the nuclear power plant outlet (NPP OL), which was considered a natural mesocosm subject to long-term hot water perturbation. Seawater in the bay comes from the Kuroshio Current in winter and South China Sea current in summer. The average sea surface temperature in this bay is 29.0 °C in summer and 24.0 °C in winter. A total of 1913 coral colonies belonging to 60 species from 16 genera and representing 7 families were sampled in 2009 and 2010 from 3 m and 7 m depths from eight sites; Wanlitung (WLT), NPP OL, Houbihu (HBH), NPP Inlet (IL), Taioshih (TS), Tanziwan (TZW), Sanjiawan (SJW), and Longken (LK), including the reef near the NPP outlet (Fig. 1). Corals sampling was authorized by Kenting National Park project #488-100-01. For every coral genus sampled, a minimum of 15 colonies were collected from each location, if available, at each designated sampling depth. At some sampled sites, it was not possible to find even one colony of some genera, while at other sites the minimum sample was 1 and maximum was 35. Hence, it was not possible to maintain uniformity in sample number due to the uneven distribution of coral genera. All samples were immediately fixed in 90% ethanol and stored until further analysis of *Symbiodinium* clades.

### DNA extraction

DNA was extracted using the high-salt DNA extraction method that was modified according to *Ferrara et al. (2006)*. Approximately 30 mg of coral were placed in 2 ml

Eppendorf tubes into which 200 µl lysis buffer (1 M Tris-Boric, 0.5 M EDTA (pH 8), 20% sodium dodecylsulfate, and 5 M NaCl) and 10 µl of proteinase K (10 mg ml$^{-1}$) were added and incubated overnight in a 60 °C water bath. After incubation, an equal volume (210 te) of NaCl (7 M) was added to the 2 ml tube and mixed gently, and the entire solution was transferred to extraction column tubes (GP$^{TM}$ Column; VIOGENE, USA) and subsequently centrifuged at 8000 $g$ for 1 min. After discarding the flow-through, 500 µf of 70% EtOH was added and centrifuged at 8000 $g$ for 5 min. This step was repeated twice. Finally, the extraction columns were transferred to new 1.5 ml Eppendorf tubes and incubated at 37 °C for 15 min. Preheated (60 °C) TE buffer (50 µl) was used for the final extraction step, and the column was centrifuged at 13 000 $g$ for 3 min. All DNA samples were kept at −20 °C until further analysis.

### Restriction fragment length polymorphism analysis of Symbiodinium clades

Initially to assess the diversity of *Symbiodinium* at clade level, DNA from 1913 coral samples were analyzed by Restriction Fragment Length Polymorphism (RFLP) technique. Both Nuclear large-subunit ribosomal (nlsr) DNA and nuclear small-subunit ribosomal (nssr) DNA were used to investigate the clade diversity. The polymerase chain reaction (PCR) amplification of nlsrDNA and nssrDNA was modified from a previously published protocol (*Chen et al., 2005*). The DNA concentration was adjusted to 30–50 ng µl$^{-1}$ for each PCR reaction, with a final concentration of 0.2 mM dNTP, 0.5 µM primer, 1× PCR buffer with 1.5 mM MgCl$_2$, and 2.5 units of *Taq* DNA polymerase (Invitrogen$^{TM}$, USA). PCR was performed by pre-denaturation at 95 °C for 1 min followed by 30 cycles of denaturation at 94 °C for 45 s, annealing at 50 °C for 45 s, and extension at 72 °C for 2 min. The final extension was performed at 72 °C for 6 min. The D1 and D2 regions of *Symbiodinium* nlsrDNA were first amplified with the primer set (D1/D2 F: 5′-CCT CAG TAA TGG CGA ATG AAC A-3′ and D1/D2 R: 5′-CCT TGG TCC GTG TTT CAA GA-3′) (*Loh et al., 2001*), and the PCR products were then characterized using restriction enzyme *Rsa* I. The nssrDNA of *Symbiodinium* was amplified using a host-excluding primer pair (ss5z: 5′-GCA GTT ATA RTT TAT TTG ATG GTY RCT GCT AC-3′ and ss3z: 5′-AGC ACT GCG TCA GTC CGA ATA ATT CAC CGG-3′), and the products were characterized using restriction enzymes *Sau3A* I and *Taq* I (*Rowan & Powers, 1991*). All enzymes used for RFLP were purchased from MBI (Fermantas, Italy). Digested nlsrDNA PCR products were separated at 150 V by vertical gel electrophoresis for 3.5 h on a 5% acrylamide gel (30% acrylamide/bis-acrylamide (37.5:1), 10× TBE buffer, 25% ammonium persulfate, and TEMED). This was done to increase the clarity of the nlsrDNA band pattern, which sometimes was not clear on the agarose gels. The digestion products of nssrDNA PCR-RFLP were separated by gel electrophoresis on a 3% agarose gel for 3.5 h at 50 V.

### DGGE analysis of Symbiodinium types

To assess the *Symbiodinium* diversity at type level, representative DNA samples from different band patterns from the RFLP analysis were picked and subjected to ribosomal internal transcribed spacer 2 region (ITS2) amplification using primers ITSintfor2: 5′-GAA TTG CAG AAC TCC GTG-3′; ITS2clamp: 5′-CGC CCG CCG CGC CCC GCG
CCC GTC CCG CGG GAT CCA TAT GCT TAA GTT CAGC GGG T-3′ (*LaJeunesse & Trench, 2000*). Each 50 μl PCR reaction consisted of 50 ng genomic DNA, 1× PCR Buffer, 2.5 mM MgCl$_2$, 0.4 mM dNTPs, 0.4 μM of each primer, and 2 units of Taq polymerase (Invitrogen, USA). PCR was run on a P × 2 thermal cycler (Thermo Scientific, USA) with touch-down PCR (*LaJeunesse, 2002*) to ensure specificity. The initial denaturing period was at 92 °C for 3 min, followed by 20 cycles of 30 s at 92 °C, and annealing conditions from 62 °C were decreased by 0.5 °C to the final annealing temperature of 52 °C for 30 s at 72 °C. Once the annealing temperature reached 52 °C, it was maintained at that level for another 20 cycles, followed by a final extension period of 10 min at 72 °C. Each PCR product was loaded onto an acrylamide denaturing gradient gel (45–80%) and then electrophoresed at 115 V for 15 h using a CBS Scientific system (Del Mar, CA, USA). Gels were stained with SYBR Green (Molecular Probes, Eugene, OR, USA) for 15 min and photographed for further analysis. Band patterns were confirmed by sequencing the cut bands from the DGGE gel.

## Coral community data

Historical coral community data for NPP OL was obtained from various published reports. Although it is difficult to compare historical data with the present data due to a change in methodology over the years, it is still possible to get an idea of how the community readjusted or redistributed over time. For 1986, we used community data previously published (*Dai, 1988*; *Dai et al., 1998*) that was obtained with quadrat sampling to estimate the percent cover of each species at NPP OL. We compared this with the data from our own 2009–2010 survey using a 20 m transect line at NPP OL. We also examined photographs taken of the coral community at NPP OL in 1986 and 1995 and compared them with 2010 photographs obtained from the same locations. Due to problems in comparing the data, it was not possible to compare them quantitatively, so we only show the pictorial comparison of how some coral genera have been replaced over time.

## Historical and present day environmental data

Seawater temperature data from 1982 to 1992 were obtained from the Taiwan Power Company's eleven-year ecological survey of waters adjacent to the nuclear power plant. Data after 1995 was obtained from different sites recorded with temperature loggers (HOBO Pendant$^{TM}$, USA).

To analyze the relationship between *Symbiodinium* diversity trend and environmental parameters, data was also obtained as time series for sea surface temperature (SST which included; average temperature, Tavg; summer average temperature, Ts; winter average temperature, Tw), *chlorophyll a* (Chl a), photosynthetically active radiation (PAR) and diffuse attenuation coefficient (Kd) over the time period 2002–2010 (at 4 km spatial resolution) from MODIS/aqua interface of the Giovanni online data system, developed and maintained by the NASA GES DISC.

For contour diagram of the seasonal seawater temperature across the Nanwan bay, seawater was collected using CTD at various sites between 2008 and 2010. The exact positions of sampling sites were located by global position system (GPS).

The measurements of temperature were carried out with CTD instrument (Sea-Bird Model 19 plus) by the EPA/ROC (Taipei) on fishing boats. The precision for temperature was ±0.05 °C. After the data output from CTD, contour maps were created using Surfer (SSG-Surfer.com, WINDOWS platform).

## Statistical analysis

The nonparametric Mann–Whitney test (without assuming a Gaussian distribution) was carried out to examine whether there were significant differences in seawater temperatures between the two depths at NPP OL and between NPP OL and different sites (WLT, MBT, STZ, and IL). To compare *Symbiodinium* clade distributions, the Chi-Square test and single-factor between-subjects ANOVA (independent samples) were used. The ANOVA results were plotted as one-way diamond mean comparisons to see the differences in *Symbiodinium* composition at different sites between 3 m and 7 m and for total *Symbiodinium* C and *Symbiodinium* D composition at NPP OL (3 m and 7 m). The horizontal dashed line in the diamond plot is the overall mean (i.e., grand mean). The line through the center of each diamond is the group mean. The top and bottom diamond vertices are the respective upper and lower 95% confidence limits (CI) about the group mean.

To understand the relationship between the environmental data, coral host and *Symbiodinium* at different sites and depths, similarity (Bray-Curtis) in the host and symbiont data (presence/absence) was analyzed using Principle Coordinate Analyses (PCO) and ANOSIM. The host and symbiont similarity matrices were analyzed for a significant relation (Spearman rho, rho ≈ 0 indicates no relation is found, rho = 1 indicates a perfect relation) using RELATE. Distance-based analysis on a linear model (distLM) was used to model the relationship between the symbiont dissimilarity data and the environmental variables. To include the host effect on the symbiont matrix, PCO1 and PCO2 of the host presence/absence data (HPCO1 and HPCO2 for continued reference) were added as covariates to the environmental data matrix in subsequent linear regression data analyses (see *Tonk et al., 2013*). In the distLM, marginal tests assessed the importance of each variable separately and a forward search of the optimal fit based on an adjusted $R^2$ was used by sequentially adding environmental variables. The data was visualized with distance based redundancy analyses (dbRDA) ordination plots. Vector overlays using the environmental data and symbiont data separately as predictor variables (drawn as multiple partial correlations) were applied to visualize the effect, strength and direction of the different variables in the ordination plots. The symbiont distributions on a clade and type level were explained with environmental data (normalized), in which the collected host information included either on a species level or genus level. All symbiont and host data was transformed to relative abundance. Analysis to compare *Symbiodinium* clade distributions was carried out using GraphPad Prism (GraphPad Software, USA) and multivariate analyses and regression analyses were performed in PRIMER-e (v6.1.13) with the PERMANOVA add-on (v1.03; *Anderson, Gorley & Clarke, 2008*).

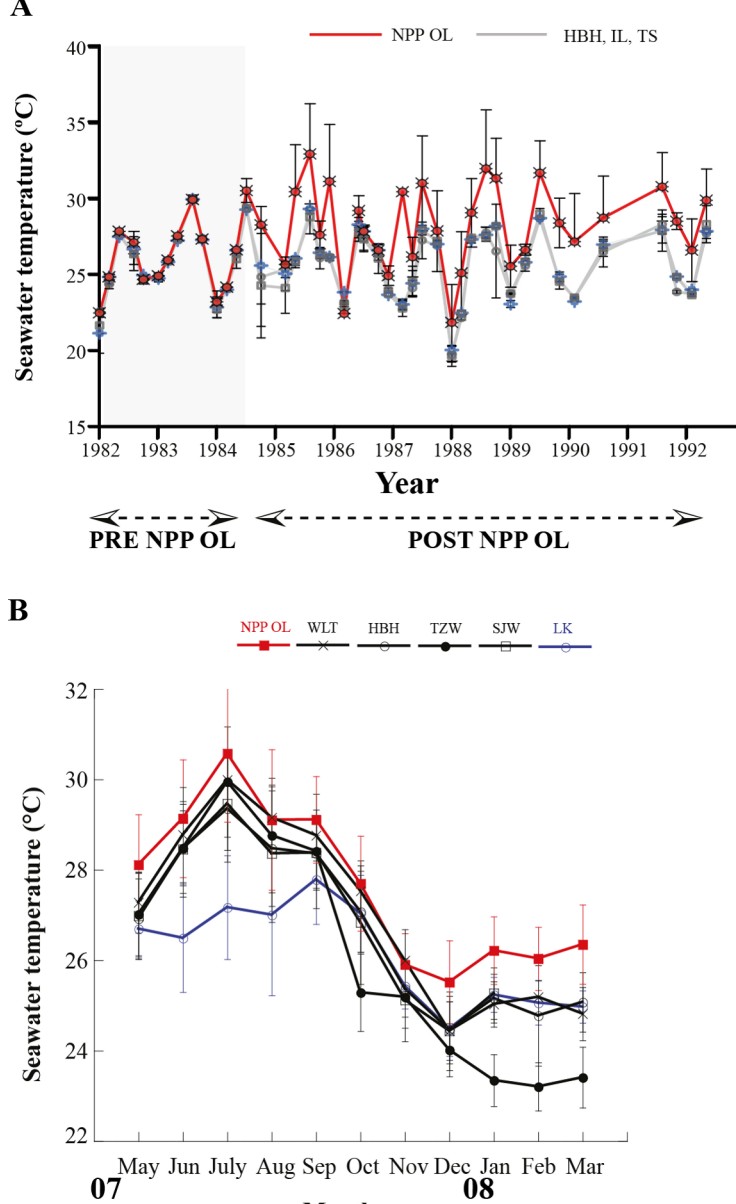

**Figure 2 Mean seawater temperatures.** NPP OL (red line) and control (grey lines) locations measured at 3 m depth from 1986 to 1992 (A). Monthly average seawater temperatures at six locations including NPP OL (B).

## RESULTS

### Seawater temperatures in Kenting

Seawater temperature data from 1982 to 1992 (Fig. 2A) shows that, prior to construction of the nuclear power plant, the average seawater temperature (26.0 °C) was the same at all locations where seawater temperature data could be obtained. However, after construction of the nuclear power plant in 1984, the average seawater temperature at NPP OL was

consistently 2.0–3.0 °C higher (Fig. 2A) than adjacent locations. Monthly and daily average seawater temperatures showed a similar trend (Fig. 2B) (also see *Keshavmurthy et al., 2012*; *Hsu et al., 2012*). NPP OL seawater temperatures at the 3 m depth were significantly higher than 7 m depths ($p < 0.0001$). The thermal effluent in NPP OL is restricted to the water column above 3 m, and the temperature differential does not influence depths below 7 m. Also, NPP OL experiences daily fluctuation of 8.0–10.0 °C (for 6 h) due to upwelling as a result of internal tides and waves generated at Luzon Strait, which directly affects the seawater present in Nanwan Bay in Kenting (*Chen, Wang & Hsing, 2004*). Differences between daily average seawater temperatures at NPP OL and two relatively distant control sites (WLT and IL) were statistically significant (Mann–Whitney test, $p < 0.0001$), whereas those between NPP OL and two closer sites (MBT and STZ) were not statistically significant (Mann–Whitney test, $p > 0.001$) (see also *Keshavmurthy et al., 2012*). Contour diagram of the seawater temperature in Nanwan showed consistent presence of hot water plume near the NPP OL irrespective of the seasons (Fig. 3).

### *Coral–Symbiodinium associations*

From the Restriction Fragment Length Polymorphism (RFLP) data, the dominant *Symbiodinium* clade associated with corals at eight sites in Kenting was *Symbiodinium* clades C (Fig. 4A). Our analysis of the *Symbiodinium* clade in coral genera at NPP OL showed that heat-tolerant *Symbiodinium* clade D was dominant in corals at the 3 m depth (Figs. 4A and 4B). Coral genera at the 7 m depth, however, were associated with *Symbiodinium* clade C at all study locations including NPP OL (Figs. 4A, 4B and 5). In contrast, *Symbiodinium* clade C was dominant at 7 m (199 out of 234 samples, 85%, Chi-square test: $X^2_{(0.01)}$:0.0001) (Figs. 4A and 4B), with only five genera having a mixed composition of *Symbiodinium* clade C and *Symbiodinium* clade D (Figs. 4A and 4B). The difference in *Symbiodinium* clade D clade between 3 m and 7 m at NPP OL was significant (one way ANOVA; $p = 0.001$). A similar significant difference was also observed in the presence of *Symbiodinium* clade C between 3 m and 7 m at NPP OL (Fig. 5).

Based on the results from ITS2 DGGE, the *Symbiodinium* types found in coral hosts belonged to type D1a (mainly found in coral hosts present in NPP OL 3 m) and types C1, C3, C21a and C15. Since we have utilized both RFLP and DGGE for analysis, to avoid confusion, throughout the text we will follow the nomenclature used for DGGE and discuss the results using DGGE ITS2 *Symbiodinium* types.

Of sixteen genera (Fig. 4), twelve genera hosted significantly more *Symbiodinium* D type D1a at NPP OL 3 m in response to thermal stress and mostly or only *Symbiodinium* types C1, C3, C21a and C15 at NPP OL 7 m (*Acropora*, *Cyphastrea*, *Goniastrea*, *Isopora*, *Platygyra*, *Favites*, *Pocillopora*, *Acanthastrea*, *Leptoria*, *Montastraea*), except for *Pavona* and *Montipora* which had *Symbiodinium* types C1 and C15 respectively as the dominant clade at both NPP OL 3 m and 7 m. The genus *Porites* was specifically associated with *Symbiodinium* type C15 at all locations, and genus *Galaxea* was associated with *Symbiodinium* type D1a at all locations and rarely hosted *Symbiodinium* type C1 (Fig. 4B, also see Fig. S1). Finally, two genera, *Seriatopora* and *Stylophora*, were associated

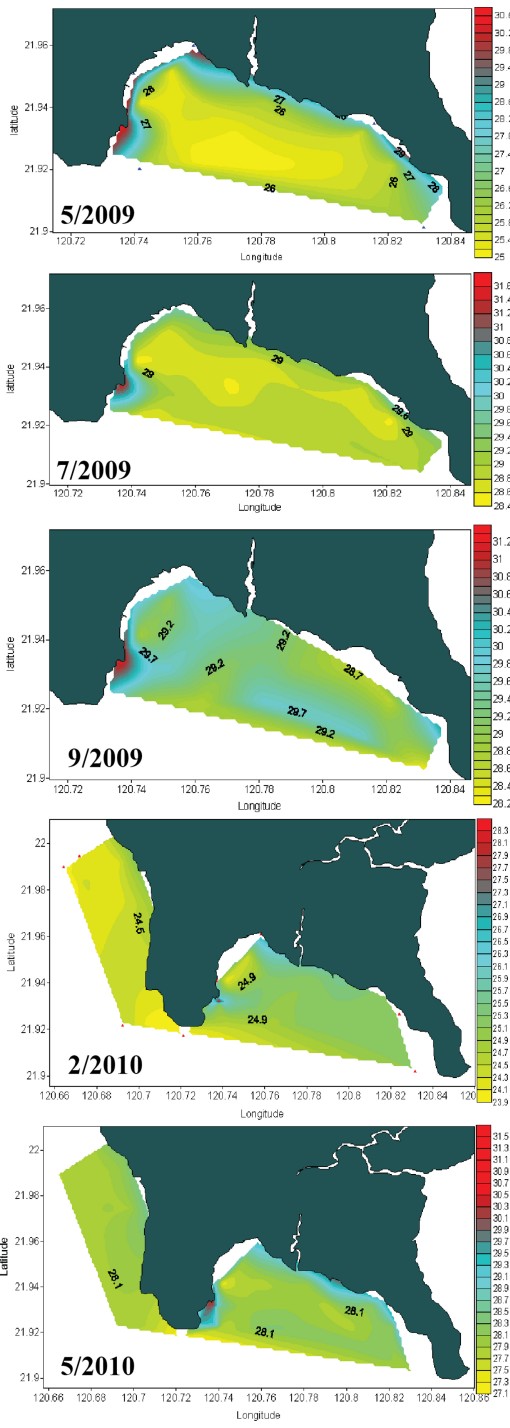

**Figure 3 Contour diagram for the seawater temperature at Nanwan.** The contour diagram of the sea surface temperature in Nanwan from May 2009 to May 2010. The red plume at the left is the constant hot water output form the nuclear power plant. The seawater near the nuclear power plant outlet are constantly hot irrespective of season.

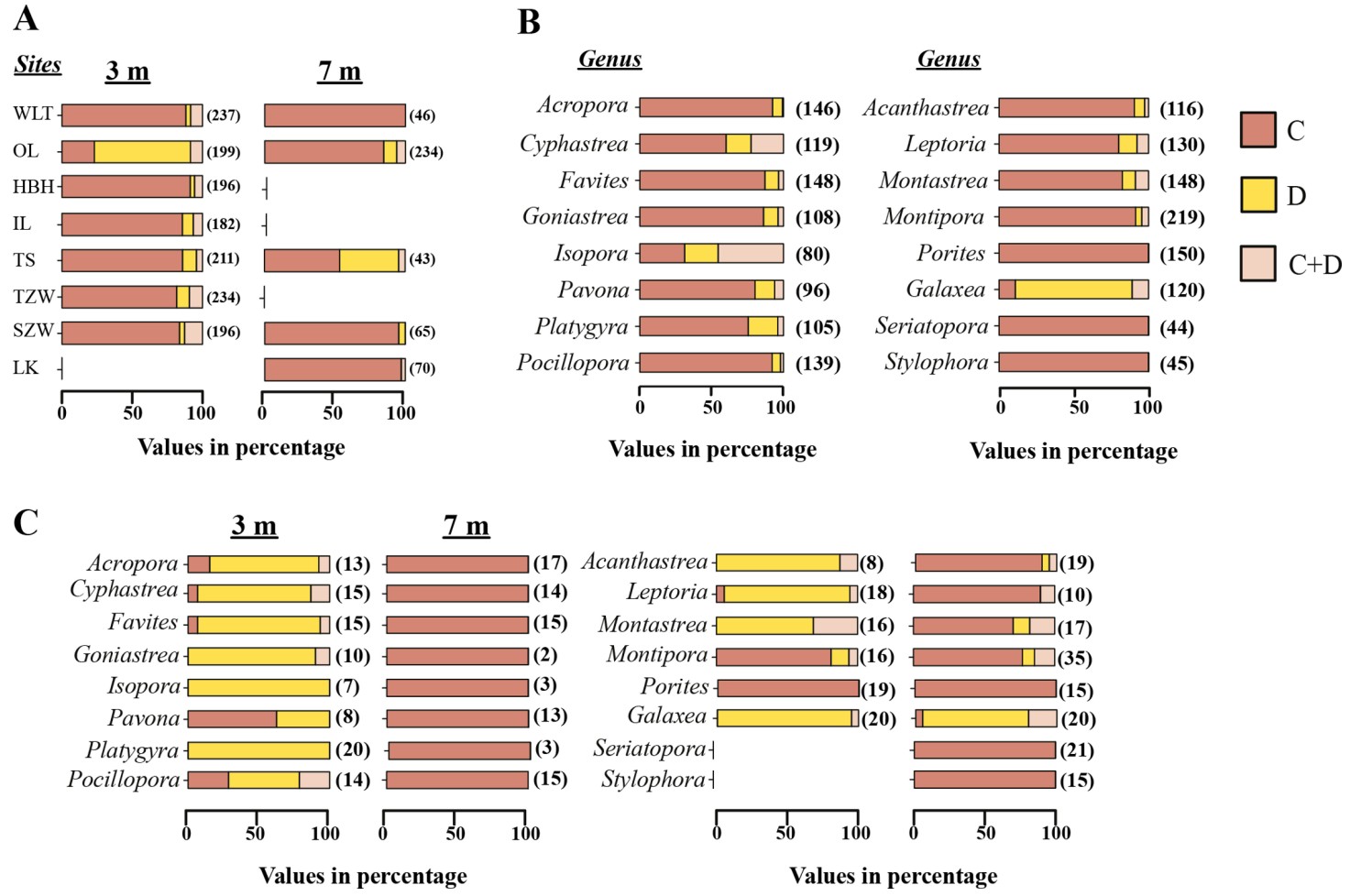

**Figure 4** ***Symbiodinium* composition in 16 coral genera sampled at 3 m and 7 m seven locations and shown separately for NPP OL.** Distribution of *Symbiodinium* clades based on the restriction length fragment polymorphism (RFLP) at 3 m and 7 m in all coral hosts at 8 sites (A). Distribution of *Symbiodinium* clades in individual genera sampled from 8 sites (B) and distribution of *Symbiodinium* clades in 16 genera in NPP OL 3 m and 7 m (C). Brown bars, *Symbiodinium* clade C; light brown bars, *Symbiodnium* clades C + D, and yellow bars, *Symbiodinium* clade D. The values in the brackets are sample numbers.

with *Symbiodinium* C type C1 and were entirely absent at the thermally stressed NPP OL site (Fig. 4C, see Fig. S1).

At sites other than NPP OL, the degree to which some genera hosted *Symbiodinium* type D1a or a combination of *Symbiodinium* type D1a and clade C types varied considerably. No *Acropora* samples were observed to do so, *Pocillopora* and *Platygyra* did so at only one other location, and other genera (*Cyphastrea, Goniastrea, Isopora, Pavona, Acanthastrea, Leptoria, Montastraea*) did so at multiple locations. However, genera *Cyphastrea, Isopora, Pavona,* and *Goniastrea* routinely hosted only *Symbiodinium* clade D1a at non-thermally stressed locations (Table S1). Species level comparisons among NPP OL (warm water influence) and WLT and SJW (sites without warm water influence) also showed that the 3 m site at NPP OL was dominated by *Symbiodinium* clade D1a, while at other sites and

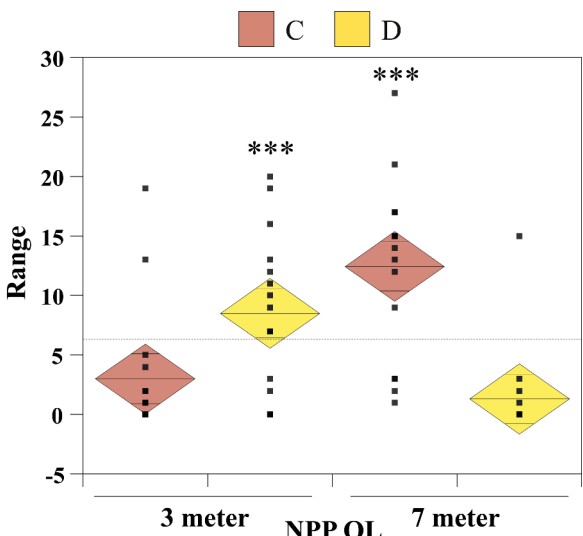

**Figure 5** One-way diamond mean plots of *Symbiodinium* clade C and *Symbiodinium* clade D. Comparison of total *Symbiodinium* clade C and *Symbiodinium* clade D composition in 16 coral genera at NPP OL (3 m and 7 m).

at NPP OL 7 m depths the dominant *Symbiodinium* belonged to clade C types (Table S1, Figs. 4C, S1A and S1B).

## Coral community

A comparison of the 1986 coral community at 3 m depths at NPP OL to the 2010 community showed a difference in the dominant species (Table 1). While *Acropora* (24.20% of total coral cover) and *Montipora* (33.69% of total coral cover) dominated the shallow waters in 1986, *Montipora* (21.70% of total coral cover) and *Galaxea* (21.20% of total coral cover) dominated the 2010 coral community. Other genera (*Favites*, *Pavona*, and *Porites*) that were <2% of the total abundance in 1986 had increased their relative abundance by 2010 (Table 1). By 2010, the presence of *Acropora* had been greatly reduced (0.70%) and *Seriatpora* and *Stylophora* were completely absent at 3 m depths at NPP OL (Figs. 6 and 7, Table 1). Results from DistLM and Bray-Curtis similarity resemblance analysis on the coral community presence or absence at different locations also showed dominance of *Galaxea* at NPP OL 3 m (Fig. 8C).

## Relationship between sites, *Symbiodinum* distributions and environmental factors

In terms of seawater temperature differences, there was a clear difference between NPP OL 3 m and other sites with NPP OL 3 m always appearing as an out group in the analysis (Figs. 8A and 8B, Table 2). Analysis also showed that several environmental parameters (Tavg—long term from 2002 to 2010 and Tsd), including the host, were responsible for driving the symbiont community. There was a clear pattern in terms of symbiont distribution (both on a cladal and type level) and NPP OL 3 m was significantly different in its symbiont composition (Fig. 8A, Table 2). The distribution of different genera among

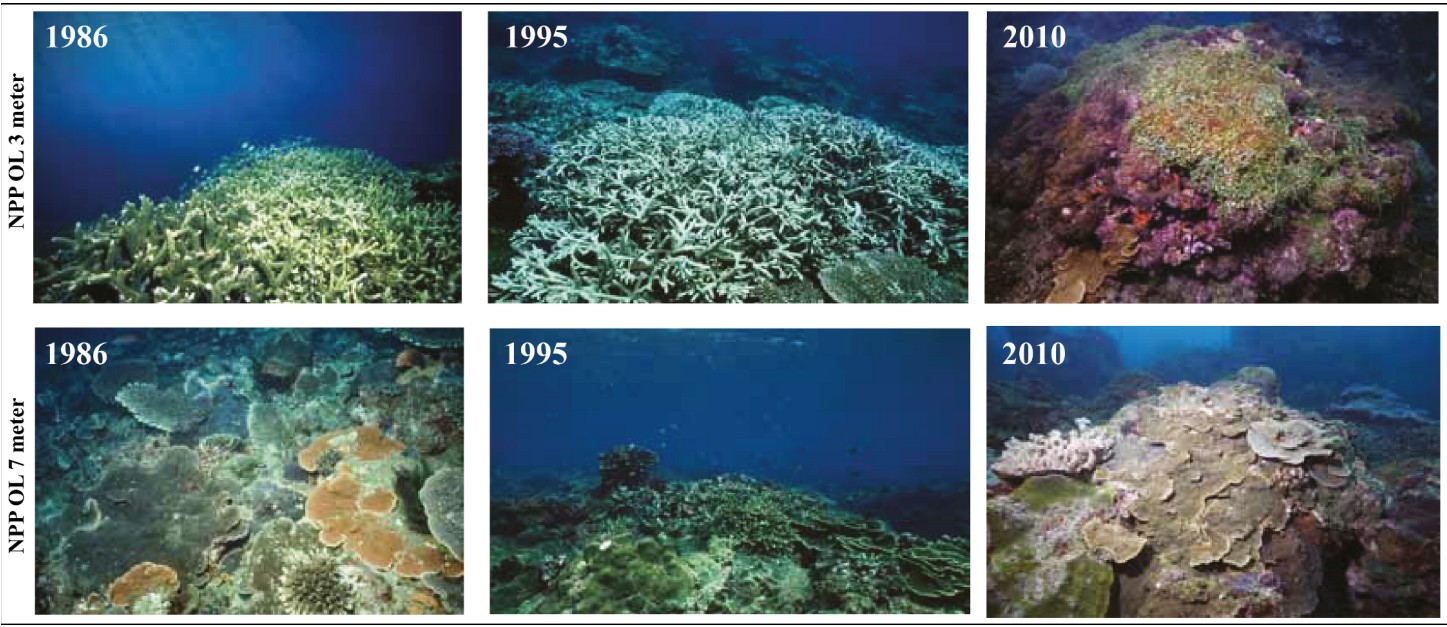

**Figure 6** Coral communities at NPP OL over time showing the condition of reefs at 3 m and 7 m in 1986, 1995 and 2010.

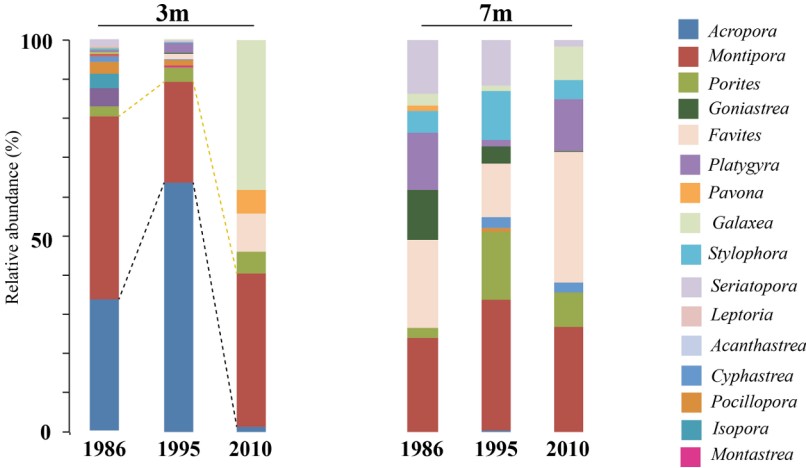

**Figure 7  Coral host composition at 3 m and 7 m NPP OL site.** Relative abundance of coral genera in 1986, 1995 and 2010 at NPP OL 3 m and 7 m. Black dotted line shows the change in *Acropora* abundance over time. Yellow dotted line shows the relative abundance of *Montipora* over time.

the sites reflected the results of the present day coral community data. Genus *Galaxea* was mainly clustered with NPP OL 3 m (Fig. 8C).

## DISCUSSION

This study observed the effect of chronically elevated seawater temperatures on the composition of coral hosts and coral-*Symbiodinium* in sixteen coral genera between NPP OL and adjacent control sites. Our results clearly showed the presence of more *Symbiodinium* type D1a at NPP OL (3 m) compared with other sites, and *Symbiodinium*

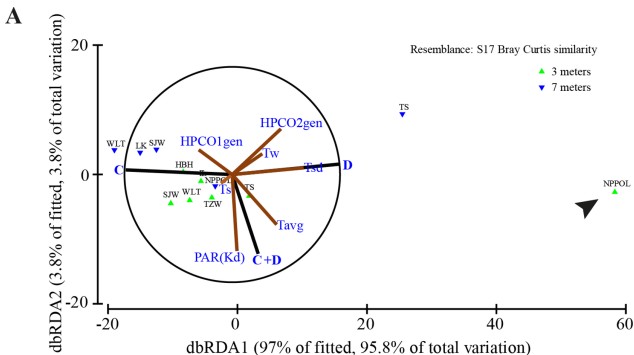

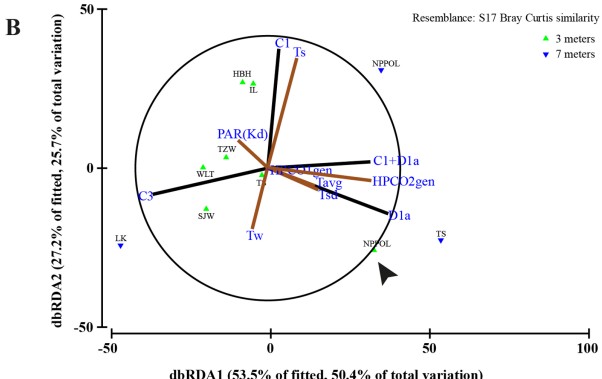

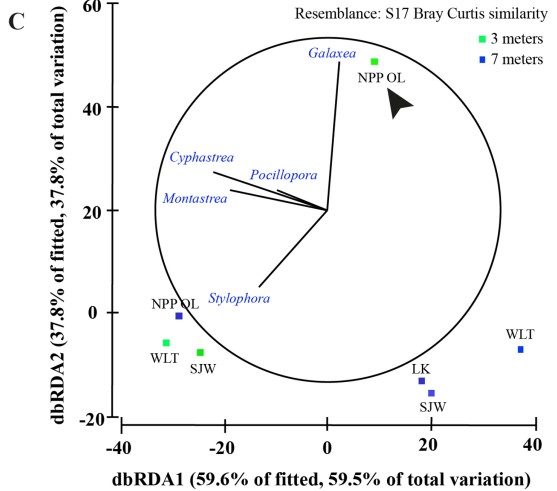

**Figure 8 Distance based RDA plots.** Environmental parameters and host genus information to *Symbiodinium* clades using all the host genera (A). Environmental parameters and host genus information to *Symbiodinium* types using the all the host genera (B) and relation between genera and sampling sites (C). Biplot projections are shown for the effect of environmental factors including host data (HPCO1 and 2) and for the occurrence of a particular genera in relation to a sampling site. 'The % of fitted' explains the percentage of the variability in the original data explained by the axis, and 'the % of total variation' indicates the percentage of variation in the fitted matrix explained by the axis. Abbreviated site names are: Wanlitung, WLT; Houbihu, HBH; NPP Inlet, IL; Taioshih, TS; Tanziwan, TZW; Sanjiawan, SJW; and Longken, LK; and NPP Outlet, NPP OL.

**Table 1** Comparison of coral assemblages between 1986 and 2010 at NPP OL location.

| Year | Major assemblages at NPP OL (3 m) | % of total cover | Minor assemblages at NPP OL (3 m) | % of total cover | Reference |
|------|-----------------------------------|------------------|-----------------------------------|------------------|-----------|
| 1986 | *Acropora* | 24.20 | *Montastrea* | 3.35 | *Dai (1988)* |
|      | *Montipora* | 33.69 | *Pocillopora* | 2.27 | |
|      | | | *Isopora* | 2.52 | |
|      | | | *Porites* | 1.84 | |
|      | | | *Seriatopora* | 1.58 | |
|      | | | *Stylophora* | 0.24 | |
|      | | | *Pavona* | 0.14 | |
| 2010 | *Galaxea* | 21.20 | *Favites* | 5.40 | This study |
|      | *Montipora* | 21.70 | *Porites* | 3.00 | |
|      | | | *Pavona* | 3.30 | |
|      | | | *Acropora* | 0.70 | |
|      | | | *Seriatopora* | Not detected | |
|      | | | *Stylophora* | Not detected | |

C types were dominant at 7 m at all locations including NPP OL (Figs. 3 and 4). Our results suggest that the present day coral host *Symbiodinium* combinations could be due to the long-term input of hot water. This can be seen by the dominance of *Symbiodinium* type D1a in the coral hosts present near the NPP OL site. We posit that the observed composition of *Symbiodinium* associated with the corals at NPP OL (3 m) might have occurred over years of acclimatization of individual hosts and *Symbiodinium* clades exposed to higher and more variable temperatures and adaptation through selection and parallel evolution of resistant host-*Symbiodinium* combinations. A reef with a higher abundance of *Symbiodinium* clade D-dominated holobionts is assumed to have a higher tolerance to thermal stress (*Ortiz, González-Rivero & Mumby, 2012*). Alternatively, the present coral-*Symbiodinium* composition at NPP OL (3 m) might be a result of the adaptive plasticity of competent coral hosts and *Symbiodinium* either separately or in combination over time. We suggest that the only plasticity that predictably enhances fitness and is most likely to facilitate adaptive evolution on ecological timescales in new environments is that which places populations close enough to a new phenotypic optimum for directional selection to act (*Ghalambor et al., 2007*). Long-term environmental standing of above average seawater temperatures at NPP OL 3 m and other factors such as nutrient input into the Nanwan might also be the reason for the present day structuring of the coral host and *Symbiodinium* diversity. However, from our analysis, there was a clear relationship between the environmental factors and distribution of *Symbiodinium* clades. The sampling sites at Nanwan, although not very separated from each other, are different in the way they are affected mainly by seawater temperature (*Hsu et al., 2012*). This is both due to the physical differences in temperature fluctuations, internal waves, upwelling and the constant out put of hot water into the NPP OL site and surrounding areas

**Table 2  Marginal tests with host genus HPCO1 and 2 and *Symbiodinium*.**

| Variable | Pseudo-F | P | % variance explained |
|---|---|---|---|
| **(a) Marginal tests with host genus HPCO1 and 2 and *Symbiodinium* type** | | | |
| PAR(Kd) | 0.83722 | 0.485 | 53.47 |
| Tw | 0.62396 | 0.681 | 27.24 |
| Ts | 1.5853 | 0.204 | 11.53 |
| Tavg | 1.252 | 0.315 | 8.28 |
| Tsd | 1.3608 | 0.305 | 4.61 |
| HPCO1gen | 0.35728 | 0.801 | −1.13 |
| HPCO2gen | 4.369 | 0.006 | −4.01 |
| **(b) Marginal tests with host species HPCO1 and 2 *Symbiodinium* type** | | | |
| PAR(Kd) | 0.83722 | 0.483 | 53.87 |
| Tw | 0.62396 | 0.637 | 29.55 |
| Ts | 1.5853 | 0.2 | 11.3 |
| Tavg | 1.252 | 0.314 | 5.26 |
| Tsd | 1.3608 | 0.27 | 3.72 |
| HPCO1sp | 1.6455 | 0.209 | −3.7 |
| HPCO2sp | 5.5322 | 0.001 | |
| **(c) Marginal tests with host genus HPCO1 and 2 and *Symbiodinium* clade** | | | |
| PAR(Kd) | 0.79276 | 0.482 | 96.98 |
| Tw | 1.7847 | 0.182 | 3.82 |
| Ts | 3.766 | 0.071 | 0.58 |
| Tavg | 4.862 | 0.063 | 0.1 |
| Tsd | 11.06 | 0.029 | 0 |
| HPCO1gen | 1.4382 | 0.203 | −0.09 |
| HPCO2gen | 1.8179 | 0.165 | −1.39 |
| **(d) Marginal tests with host genus HPCO1 and 2 and *Symbiodinium* clade** | | | |
| PAR(Kd) | 0.79276 | 0.473 | 96.86 |
| Tw | 1.7847 | 0.197 | 3.88 |
| Ts | 3.766 | 0.055 | 0.59 |
| Tavg | 4.862 | 0.066 | 0.14 |
| Tsd | 11.06 | 0.031 | 0 |
| HPCO1sp | 0.3212 | 0.659 | −0.08 |
| HPCO2sp | 4.9074 | 0.057 | −1.4 |

(*Hsu et al., 2012*; *Keshavmurthy et al., 2012*). Such factors might have induced the present day *Symbiodinium* distribution in those coral hosts present at different sites in Nanwan.

In studies conducted to date, the impact of thermally tolerant *Symbiodinium* has largely been documented at the colony scale, and the consequences at the population and community level within an ecosystem context are unknown (*Ortiz, González-Rivero & Mumby, 2012*). We show that at community level, *Symbiodinium* type D1a was dominant in 12 of 16 genera living at the 3 m depth at NPP OL (Figs. 4C and S1), whereas the same 12 genera at the 7 m depth at NPP OL were associated only with *Symbiodinium* types C1, C3 and C21 (Figs. 4C and S1). The depth-related stratification in the clade association at

NPP OL might be due to decades of constant seawater temperature elevation (2.0–3.0 °C). At the generic level, most of the corals at NPP OL 3 m hosted solely *Symbiodinium* type D1a. Two genera *Montipora* and *Porites* associated mostly or solely with *Symbiodinium* type C15. While the strategy of dominant 12 genera was their capacity to associate with a stress-tolerant *Symbiodinium* clade, the strategy followed by *Porites* could be a result of a stress-resistant mechanism. Genus *Galaxea* was abundant and associated with *Symbiodinium* type D1a at almost every site it was collected, indicating that *Galaxea* is able to tolerate disturbances and thrive in perturbed environments. From the above observations we believe that such differences, despite constant warm water and a large daily fluctuation in the seawater temperature (*Chen, Wang & Hsing, 2004*), is due to the ability of coral–*Symbiodinium* combinations to thrive or the host itself to be able to survive in such conditions. And because of the inability of some genera to host *Symbiodinium* type D1a or any other resistant *Symbiodinium* combination, *Stylophora* and *Seriatopora* were absent from 3 m at NPP OL.

Many previous studies (see the references in the Introduction) have shown that the shuffling of *Symbiodinium* and/or host resistance mechanisms can confer resistance in corals to environmental stress. From the observations in this study and a recent study (*Keshavmurthy et al., 2012*), we suggest there is the possibility of a strategy shift (*Done, 1999*) in corals present in NPP OL or other sites, although we could not demonstrate the shuffling of *Symbiodinium* in corals *per se*. NPP OL populations at 3 m would represent stressed populations that have evoked a dominance of stress tolerant *Symbiodinium* type D1a as one option during their adjustment to the upper limit of their thermal range. There are at least two other alternative explanations, however. First, *Symbiodinium* type D1a may be an opportunistic type (see *Stat & Gates, 2011*) that occupies compromised coral hosts, resulting in a less than optimal symbiosis and reduced rates of host growth. In this case, NPP OL corals at 3 m are stressed and populated by less optimal varieties of *Symbiodinium*. Although corals associated with *Symbiodinium* type D1a may benefit in the short term by surviving bleaching, there are clearly trade-offs in terms of fitness that may have major implications for the long-term growth and survival of coral reefs (*Stat & Gates, 2011*; also see *Ortiz, González-Rivero & Mumby, 2012*), thereby negatively affecting the competitive ability of corals (*Baker et al., 2013*). For example, the association of *A. tenuis* juveniles with *Symbiodinium* D resulted in higher metabolic costs and lower physiological tolerances (*Abrego et al., 2008*). Second, new host-*Symbiodinium* combinations could arise as a result of directional selection where all combinations of host-*Symbiodinium* arrive in the shallows at NPP OL but only those with significant associations with *Symbiodinium* type D1a survive. The latter explanation would be a consequence of the fact that NPP OL is open to populations not experiencing the same levels of thermal stress, and hence the resulting composition of *Symbiodinium* is a consequence of active selection for the *Symbiodinium* that occupy these coral genera and not shuffling *per se*. However, *Baker et al. (2013)* have posited that acute disturbances or long-term environmental conditions are not sufficient to explain the composition of symbiont communities and their dynamics in any given location. The trend seen in the symbiont composition can be more explained

by physical factors such as daily seawater temperature fluctuations as a result of tidal oscillations or internal waves induced upwelling, which act as acute and chronic stressors. This can be one explanation for the presence of dominant *Symbiodinium* type D1a in the coral hosts in NPP OL 3 m. Constant hot water output and daily temperature fluctuations up to 10 °C (*Chen, Wang & Hsing, 2004*) in the NPP OL site might have allowed *Symbiodinium* type D1a to be maintained at relatively high levels at NPP OL 3 m. This also shows that *Symbiodinium* type D1a isn't necessarily always opportunistic, thermally tolerant and transient during stress conditions, but also can be capable of long-term persistence on reefs with favorable conditions (*Baker et al., 2013*). All the hypotheses that we have put forth above should be considered as alternatives to explain the trend of existing coral–*Symbiodinium* associations at warm water-affected locations at NPP OL.

In the case of coral hosts, there were considerable changes in the types of dominant coral at 3 m while the composition at 7 m at NPP OL did not change over the years (Table 1, Figs. 6 and 7). The present day framework of the NPP OL community could be due to a strategy shift (*Done, 1999*) between *Symbiodinium* and coral stages, where both transient and continual states exist in the composition of associated *Symbiodinium* clades and coral genera. In the case of Kenting reefs, strategy shifting exists not only between the two water depths at NPP OL but also between the sites (see *Kuo et al., 2012*). From the long term satellite seawater temperature data, the averages of temperature at NPP OL 3 m is similar to the WLT site, however, the analysis results showed otherwise, with TS 7 m as more similar to NPP OL 3 m indicating some other factors in addition to the above mentioned environmental factors influencing the patterns seen in the host as well as *Symbiodinium* distributions. But the distLP analysis suggested that the environmental data more than sufficiently explains the variation seen in the symbiont community. The other explanation for the difference seen in the symbiont distribution might be the differences in the host distribution among the sites. But our sampling design made sure to sample hosts as uniformly as possible at all sites, except in such cases where a particular host was absent and could not be sampled.

The differentiation between coral and *Symbiodinium* populations among the different sites at Kenting might have been a result of populations developing tolerance to along-shore gradients of environmental factors, including temperature and pollution. At scales of less than 1 km, differentiation can occur over short, vertical stress gradients along horizontal gradients of wave exposure covering a few hundred meters; (*Sanford & Kelly, 2011*). Variations in seawater temperatures and anthropogenic changes in Kenting might have led to adaptive divergence in physiological traits among populations distributed across a variety of scales. Studies have shown that human and natural disturbances are similar at all the sites represented in this study (*Meng et al., 2008*; *Kuo et al., 2012*). Apart from long-term hot water perturbations at NPP OL, other areas are not much different in terms of exposure to stress at Kenting. While coral communities at other Kenting sites show considerable changes over time (*Kuo et al., 2012*), the coral presence at NPP OL is still diverse and there is no evidence of a shift in the community from corals to

algae. Studies have suggested that historical effects play an important role in determining the fate of individuals, populations, and communities (from *Marshall & Baird, 2000*). In case of corals, a historical thermal exposure can influence their thermal tolerance (see *Howells et al., 2013*), and this could be through acclimatization and selection for tolerant genotypes. Also, over longer periods of time (between generations), selection may drive heritable changes in the mean phenotype, but in long-lived individuals such as corals the genetic adaptation might be slow and occur over decadal time scales (*van Oppen et al., 2011*; *Császár et al., 2010*; *Hoegh-Guldberg et al., 2007*; *Aitken et al., 2008*). However, genotypic variation among individuals allows the species to persist through the expansion of genotypes better suited for a new climate (*Richter et al., 2012*; also see *Kramer et al., 2010*). *van Oppen et al. (2011)* suggested that somatic mutation during asexual reproduction could aid corals in evolution and adaptation (see also *Fautin, 1997*; *Buddemeier, Fautin & Ware, 1997*). Evolution within mitotic cell lineages, both in the coral host and *Symbiodinium* (*Correa & Baker, 2010*), might play a role in the adaptation of corals to climate change.

## CONCLUSIONS

Our work and a previous study (*Berkelmans & van Oppen, 2006*) show that even if several coral species could withstand temperature increases beyond 1.0–1.5 °C, the surviving coral species may not be sufficient to maintain healthy reefs. Due to long-term acclimatization, community level resistance to perturbations (in our case, seawater temperature) is possible through association with stress resistant *Symbiodinum* type D1a. Acclimatization is an important strategy by which individuals can adjust phenotype to perform more optimally under changed environmental conditions.

Studies have shown that in coral reefs the process of local adaptation and acclimatization to high average temperatures and recurrent thermal stress is possible (*Rowan et al., 1997*; *Marshall & Baird, 2000*; *Brown et al., 2002*; *Howells et al., 2012*; *Howells et al., 2013*). There are limits to acclimatization, however, which are set by tradeoffs at various structural and functional levels that ultimately constrain the width of the thermal range of a given species (see *Doney et al., 2012*). These constraints highlight the limits of acclimatization, which ultimately results in shifting community composition as tolerance thresholds in the more vulnerable species of a given community are exceeded. Laboratory research has shown that genetic variation for plasticity exists (*Barshis et al., 2010*; *Császár et al., 2010*) and heritable plasticity can respond to artificial selection (*Nussey et al., 2005*). Given that corals are exposed to long-term anthropogenically driven environmental change (*Hoegh-Guldberg, 1999*; *Hoegh-Guldberg et al., 2007*), it is imperative to obtain a better understanding of how natural selection acts on plasticity under altered levels of environmental variation in the wild (*Nussey et al., 2005*). Understanding the limits of these combinations will allow us to understand why some corals are 'winners' and others are 'losers' during the early stages of rapid anthropogenic climate change (see *Loya et al., 2001*). To some extent, the development of an optimal combination of coral hosts (from around 800 species) and *Symbiodinium* (over 100 distinct genetic varieties

from 4 major clades) is a complex (*Goulet, 2006*) and stochastic process somewhat like a lottery (*Jones, 2008*). *Parmesan & Yohe (2003)* provided evidence of ecosystems altered by climate change. Our results suggest that corals assemblages and their symbionts exposed to warmer waters are already undergoing alteration. Our data also suggest that not all coral species in a given community have the ability to acclimatize to survive in warmer water. Current evidence suggests that natural adaptation within coral populations is unlikely to occur quickly enough to keep up with rapid changes in ocean temperatures, and although shuffling *Symbiodinium* clades could play an important role in extending the physiological performance of a coral species, this might not be the case at the community level, resulting in the loss of some species over time (Fig. 6). If the present trend of ocean warming and change continues, we might be looking at unsustainable restructuring of coral assemblages in a given coral community (Fig. 6), which in turn would cause irreversible damage to coral reef ecosystems.

## ACKNOWLEDGEMENTS

We thank Jay Wei for field assistance and Chen-Ju Lai for *Symbiodinium* phylotyping. We also thank all 3 anonymous reviewers and Dr. Eugenia Sampayo for their comments and suggestions in improving this manuscript and for Dr. Sampayo's help in the data analysis. This is Coral Reef Evolutionary Ecology and Genetics Group, Biodiversity Research Centre, Academia Sinica contribution No. 110.

### Funding

This research is supported by Academia Sinica (AS-97-TP-B01), National Science Council (NSC 98-2321-B-001-024-MY3), and Kenting National Park (488-100-01) to Chaolun Chen. Shashank Keshavmurthy is supported by postdoctoral fellowship from Academia Sinica (2013–2014). The funders had no role in study design, data collection and analysis, decision to publish, or preparation of the manuscript.

### Grant Disclosures

The following grant information was disclosed by the authors:
Kenting National Park: 488-100-01.
Academia Sinica, Taiwan: AS-97-TP-B01.
National Science Council, Taiwan: NSC 98-2321-B-001-024-MY3.

### Competing Interests

The authors declare there are no competing interests. Dr. Pei-Jie Meng is an employee of the National Museum of Marine Biology/Aquarium.

### Author Contributions

- Shashank Keshavmurthy conceived and designed the experiments, performed the experiments, analyzed the data, wrote the paper, prepared figures and/or tables, reviewed drafts of the paper.

- Pei-Jie Meng analyzed the data, contributed reagents/materials/analysis tools.
- Jih-Terng Wang conceived and designed the experiments, contributed reagents/materials/analysis tools, reviewed drafts of the paper.
- Chao-Yang Kuo performed the experiments, analyzed the data.
- Sung-Yin Yang performed the experiments, reviewed drafts of the paper.
- Chia-Min Hsu and Chai-Hsia Gan performed the experiments.
- Chang-Feng Dai contributed reagents/materials/analysis tools.
- Chaolun Allen Chen conceived and designed the experiments, contributed reagents/materials/analysis tools, wrote the paper, reviewed drafts of the paper.

### Field Study Permissions

The following information was supplied relating to ethical approvals (i.e., approving body and any reference numbers):

Kenting National Park, Taiwan (488-100-01) and Tai-Power Plant.

### Supplemental information

Supplemental information for this article can be found online at http://dx.doi.org/10.7717/peerj.327.

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
