# Peer review of "Can resistant coral-Symbiodinium associations enable coral communities to survive climate change? A study of a site exposed to long-term hot water input"

_PeerJ, doi:10.7717/peerj.327_

## Round 0.1 · original submission · Major Revisions

This version is much improved but still requires major revision. Also, please be careful how you interpret your findings as they relate to coral responses to climate change in general. Don't overreach.

·

Basic reporting

See general comments

Experimental design

See general comments

Validity of the findings

See general comments

Additional comments

The manuscript by Keshavmurthy and coworkers presents a case study for the long-term effects of continuously increased thermal conditions on a coral reef community. The manuscript if fairly well written but some sentences are difficult to read as they become lengthy. I also find that much of the manuscript is filled with very careful statements such as ‘likely’, ‘suggest’, ‘potential’ or with acknowledgments of inferiority of the data. As it stands this may well represent one of the largest and most comprehensive data sets whereby the authors have control ‘nearby’ sites against which changes in community can be gauged. That said, I feel that the data analyses itself is quite limited especially given the large amount of information present. I would suggest the authors expand their analyses and place less emphasis on the processes by which change has occurred but rather on the importance of the differences between the communities resulting from a thermal regime shift.

Two examples of where a change in emphasis might be useful are found in the introduction and discussion:
- The paragraph from line 49 to 87 can be largely condensed or integrated (briefly) into the following one. I feel the section on adaptation, selection etc is more suited to the discussion. The underlying rationale for the study is quite straightforward and the introduction is lengthy and detracts from the main objective: The authors are interested in comparing coral communities and their symbionts along a naturally occurring thermal gradient. They wish to use this data to predict how coral communities might change in response to climate change and whether the symbionts play a role in this.

- The first paragraph of the discussion, ln 310-311. The statement is largely irrelevant if the authors show that long-standing environmental conditions, or a change therein, shape the community composition of both corals and their symbionts. Although it would be fantastic to know what the symbiosis were pre-construction of the NPP, the question here is a basic ecological/evolutionary one and considering the authors are not interested or in reality can not investigate the mechanism underlying community change. As such, the potential mechanisms can be discussed (briefly) in the discussion but I would suggest rather to focus on the outcomes of imposed environmental change, and in that respect it is not necessary to have pre-NPP data. In fact, the authors not only back their results with coral community control data pre-construction of the NPP as well as the analyses of nearby control sites. These sites should not only have similar evolutionary history and undergo similar environmental regimes, so that a deviation therefrom can directly be attributed by a change in local conditions. The differentiation of the NPP sites with nearby sites and even at the same NPP sites but with depth are most likely the result of long standing environmental regimes, especially if the authors can show that coral community composition at the NPP location was historically more similar to nearby control sites. That said, if the data is analysed more extensively it would make a considerable contribution due to the combination of both present-day and historical (i.e. past ~30 years) community composition data. It would further allow the authors to draw conclusions on potential shifts in community assemblages as a result of changes in thermal regimes.

Environmental data
- it would be nice to show a map of the effluent plume and the thermal gradient running along the various sample locations.
- Is there a comparison between the two thermal data collection methods? If there is overlap you should probably compare the two to make sure there are not discrepancies that may influence the temperature change over time. It would also be nice to include the data post 1995 up until the present.
- If data is available it would be useful to calculate long term monthly averages as opposed to months only in 2007-2008. The authors report that there is logger data from 1995 onwards but otherwise satellite data or source data from the NPP could be used to get long-term data. Perhaps to make the data more intuitive, the authors could combine a map with the outflow thermal gradient plume with a ‘heat map’ for the long term average summer/winter temperatures.

Host and Symbiodinium data analyses
- Differentiate between clade and type more consistently. In reality the classification to the clade level is may not be totally relevant here given the different characteristics of specific types within those clades. Although the clade D lineage of Symbiodinium is generally referred to as heat tolerant there are also D types that are clearly not. The type referred to here is D1a specifically, which is shown to be mainly opportunistic, thermally tolerant and transient during stress conditions. On the other hand Lajeunesse et al. 2010 also show a variety of other D-types, but that are host specific, in chronically warmer areas. In a similar fashion, not all C-types are sensitive, examples are C15 in Porites or Montipora or C8a in Stylophora.

- why are digested 28S rDNA products run on acrylamide gels? Were the fragments so similar in size that agarose gel electrophoresis could not separate them? Why did the authors do RFLP of 18S rDNA, acrylamide electrophoresis of 28S rDNA as well as ITS2-DGGE? The first two seem redundant as they would only distinguish to the clade level. This is not to say there is anything incorrect with these methodologies, I am just wondering why the redundancy and whether there is additional informative information here (for example from the 28S rDNA analyses). Did the authors not find any uncharacterised/new types?

- Were all 1913 samples analysed to the ‘type’ level, and if so why is this information not presented more comprehensively? The data analyses could include multivariate community analyses of between site and between depth comparisons (likely show that the deep near the NPP are more similar to locations distant from the NPP). In addition a regression between temperature regime and the relative abundance of D versus C in corals occurring at all sites could make a really nice addition to the dataset here (keeping in mind that the regression may be influenced by tolerant C-types identified here).

- It seems the authors have data on coral community assemblages from 1981. Even if the data is restricted to two time-points, i.e. prior to the construction of the NPP and the presented survey, it would make an valuable addition if they could show a shift in coral community composition over time. While the data is shown in a table as percentages, I think it would be worthwhile exploring the data with the use of multivariate statistics to show similarity between sites and estimate the contribution of Symbiodinium type as well as temperature to the coral community composition and change over time. The specific symbiont community comparison between and within sites as well as the host community should receive more attention, and again can be analysed using multivariate statistics on a community level rather than chi-square statistics.

Ln 76-77 - where is the data to back up this statement as none of the referenced studies investigated the symbiont type.

Ln 272 – it’s not clear what the distinctions of ‘weedy’, ‘winner’, ‘loser’, or ‘competent’ are based on. I would suggest providing the categorical delimitations for these categories. Based on what I can get from the discussion these are somewhat loosely based on the presence of the species (i.e. losers), but also on the association with its symbiont. In that regard, why is a host (Galaxea) that always associates with D categorised as ‘weedy’ while one (Porites) that associates with C15 categorised as a winner if both occur at all sites. It is likely that these are also patterns influenced by host-specificity, and, in this particular case, these are both symbionts that fare well under higher thermal regimes.

Reviewer 2 ·

Basic reporting

The revised paper by Shashank and colleagues is improved over the previous submission. However, in this reviewer's opinion, problems remain with the presentation of the Symbiodinium diversity data. The authors seem to be of the opinion that clade level differences are the only important distinctions in Symbiodinium diversity and that lower level variability can be dismissed as population or sub-species level information that does not provide insight into the physiological or adaptive responses of corals to prolonged thermal stress. This starts in the abstract and continues throughout the paper. While the authors are entitled to that position (and others in the field do hold this view), in my opinion it is inconsistent with a large and rich literature on Symbiodinium diversity/physiology/ecology and is therefore wrong. In my last review I encouraged the authors to review some of the papers on this subject and revise the manuscript accordingly. While some new literature has been added and the most problematic statements have been scrubbed, the general perspective of the clade as the important unit of diversity remains. Thus my recommendation to the authors is to revise their thinking, delve into recent papers - particularly those authored by Todd LaJeunesse and his colleagues - and revise the text some more.

Experimental design

During the previous round of review, the issue of chronic thermal stress vs. perturbations from the mean was raised by one of the reviewers. Some discussion of this issue and the limits of the study design should be added to the discussion. The note from the author is overselling the applicability and significance of the study and the other reviewer was right to bring the limits of the approach to the authors attention.

Validity of the findings

I generally feel that the findings of the study are valid and the data are sound, but I disagree with the framing and presentation of the Symbiodinium diversity data.

---

## Round 0.2 · accepted · Accept

The paper is much improved and now ready for publication